# Changing the IgE Binding Capacity of Tropomyosin in Shrimp through Structural Modification Induced by Cold Plasma and Glycation Treatment

**DOI:** 10.3390/foods12010206

**Published:** 2023-01-03

**Authors:** Feng-Qi Wang, Jun-Hu Cheng, Kevin M. Keener

**Affiliations:** 1School of Food Science and Engineering, South China University of Technology, Guangzhou 510641, China; 2Academy of Contemporary Food Engineering, South China University of Technology, Guangzhou Higher Education Mega Centre, Guangzhou 510006, China; 3School of Engineering, University of Guelph, 50 Stone Road East, Guelph, ON NIG 2W1, Canada

**Keywords:** cold plasma, glycation, tropomyosin, IgE binding ability, conformation

## Abstract

Tropomyosin (TM) is the major allergen of shrimp (*Penaeus chinensis*). Previous studies showed that separate cold plasma or glycation have their drawback in reducing allergenicity of TM, including effectiveness and reliability. In the current study, a new processing combining cold plasma (CP) and glycation was proposed and its effect on changing IgE binding capacity of TM from shrimp was investigated. Obtained results showed the IgE binding capacity of TM was reduced by up to 40% after CP (dielectric barrier discharge, 60 kV, 1.0 A) combined with glycation treatment (4 h, 80 °C), compared with the less than 5% reduction after single CP or glycation treatment. Notably, in contrast to the general way of CP prompting glycation, this study devised a new mode of glycation with ribose after CP pretreatment. The structural changes of TM were explored to explain the decreased IgE binding reactivity. The results of multi-spectroscopies showed that the secondary and tertiary structures of TM were further destroyed after combined treatment, including the transformation of 50% α-helix to β-sheet and random coils, the modification and exposure of aromatic amino acids, and the increase of surface hydrophobicity. The morphology analysis using atomic force microscope revealed that the combined processing made the distribution of TM particles tend to disperse circularly, while it would aggregate after either processing treatment alone. These findings confirmed the unfolding and reaggregation of TM during combined processing treatment, which may result in the remarkable reduction of IgE binding ability. Therefore, the processing of CP pretreatment combined with glycation has the potential to reduce or even eliminate the allergenicity of seafood.

## 1. Introduction

The amount of seafood consumption has risen remarkably in recent years for its unique palate and great nutritional value [1,2,3,4]. However, seafood has always been regarded as allergenic food, especially shrimp, crab, scallops, and fish [5,6]. It has become an obstacle to the development of the seafood industry and posed a threat to health of consumers. Tropomyosin (TM) is the main cause of seafood allergy with a molecular weight (MW) of 34~38 kDa [7]. Due to its thermal stability, traditional heat processing techniques have an insignificant effect on reducing its allergenicity, confusing researchers for a long time [8,9]. In recent years, several new techniques have been adopted to tackle TM of crustaceans and they obtained positive results in terms of reducing allergenicity of seafood allergens [10,11,12].

Glycation is the reaction between amino residues (α- or ε-NH_2_) of proteins and carbonyl of reducing sugars. Maillard reaction (MR) is one type of it and is commonly used to modify proteins in food industry [13]. It was confirmed that MR with ribose for 12 h could reduce allergenicity of TM by 60% [14]. This may be attributed to the destruction of allergenic epitopes, as it led to changes in allergenicity. However, glycation may also generate new epitopes and increase allergenicity, which depends on reaction conditions [15,16]. In addition to such uncertainty, the demands of long reaction time and high temperature also limit the application of glycation.

Non-thermal processing technologies have been applied in food processing recently, such as high hydrostatic pressure, ultrasound, ultraviolet, radiation, and cold plasma (CP), and they have the ability to reduce allergenicity of food allergens due to their unique properties [6]. For instance, IgE and IgG binding capacity of TM of squid (*Todarodes pacificus*) reduced by around 25% and 60%, respectively, after high hydrostatic pressure treatment (400 MPa, 20 min, 20 °C) [17]. In the meantime, Li et al. [18] found that immunoreactivity of shrimp (*Penaeus vannamei*) decreased after ultrasound treatment (30 kHz, 800 W, 30 min). CP, a novel non-thermal processing technique, has been gradually used in food preservation and sterilization [19,20,21,22,23]. It contains or indirectly produces lots of high-energy electrons, free radicals, reactive oxygen species (ROS), and reactive nitrogen species (RNS) [19]. With the ability of physical and chemical oxidation, it has been believed to have great potential in modifying proteins and reducing their allergenicity [24,25]. It was reported that the activity of the horseradish peroxidase was decreased after CP treatment due to the destruction of its secondary and tertiary structures [26]. Regarding the reduction of food allergenicity, previous studies showed that it had a positive effect on peanuts, soybean, milk, and wheat allergens [27,28,29,30]. For TM of crustaceans, Shriver [31] found a 10% reduction of IgE binding ability after dielectric barrier discharge CP treatment, while there was no reduction of IgG binding ability. Ekezie et al. [32] also found that cold argon plasma jet induced a serious destructive effect on conformation of TM, including secondary and tertiary structures. However, it did not show a significant reduction in allergenicity of TM, probably resulting from few conformational epitopes of TM of crustaceans [33].

Such severe seafood allergy is a big challenge, which is waiting urgently for an effective solution. Yu et al. [34] reported that CP increased the degree of glycation to improve the solubility of peanut protein isolate. It was said that the production of radicals could promote glycation. Moreover, due to the modification on chitosan films by CP treatment, their wettability and surface free energy increased and became easy to adhere zein [35]. Thus, the approach of combining CP with glycation has the potential to tackle seafood allergens. For one thing, the modification on structure of proteins induced by CP may be beneficial to the extent of glycation. For another, CP combined with glycation would disrupt linear and conformational epitopes simultaneously, resulting in a great reduction in allergenicity.

Therefore, the purpose of this study was to build up a new processing method to reduce allergenicity of TM of shrimp. Concretely, the method of CP combined with glycation was worth consideration. For this purpose, this study investigated the effect of the combination of CP and glycation on IgE binding activity with TM from the shrimp (*Penaeus chinensis*). Based on its conformational changes, the possible mechanism of CP combined with glycation on allergenicity of TM was established. Multi-spectroscopic methods and atomic force microscope (AFM) three-dimensional (3D) images were used to determine and characterize the conformational changes of TM.

## 2. Materials and Methods

### 2.1. Materials

Shrimp was purchased from Pupumail (Guangzhou, China). Horseradish peroxidase (HRP)-conjugated anti-human IgE antibodies (1 μg/μL) were obtained from Sino Biological Inc. (Beijing, China). The 3, 3′, 5, 5′-tetramethylbenzidine (TMB) was purchased from Beijing Solarbio Science & Technology Co., Ltd. (Beijing, China). Bovine serum albumin (BSA) was purchased from Sangon Biotech Co., Ltd. (Shanghai, China). The precast gels (Tris-Glycine, 4~20%) and the molecular weight protein standard (Prestained color protein Ladder) were bought from Beyotime Biotechnology (Shanghai, China). Serums of shrimp allergic patients were purchased from PlasmaLab International (Everett, WA, USA). They were selected based on specific shrimp TM IgE levels > 0.35 kU/L (accessed by ImmunoCAP system), which were defined as positive (shown in Table 1). Meanwhile, they all have a documented history of shrimp allergy with allergic symptoms. None of the recruited subjects had any other forms of allergies. All serums were mixed equally and used as one sample in the ELISA analysis. The serums were stored at −80 °C until analysis.

### 2.2. Extraction and Preparation of TM

TM of shrimp was prepared according to the method of Xu et al. [36] with some modifications. Briefly, the minced muscles were homogenized (1:10, *w/v*) in a buffer containing 50 mM KCl and 2 mM NaHCO_3_ for 4 h. The mixture was centrifuged at 14,000× *g* for 10 min, and the precipitates were transferred into the above same buffer for another 4 h homogenization. After centrifuging, the precipitates left were washed in 10-fold of cold 95% ethyl alcohol for three consecutive times, and the resulting residues were washed with cold acetone (v:v = 1:1) and dried in a fume cupboard. The protein powder was extracted by a buffer (1 M KCl, 0.5 mM dithiothreitol (DTT), and 20 mM Tris-HCl, pH 7.5) and stirred overnight. Subsequently, the solution was centrifuged at 14,000× *g* for 10 min and retained the supernatant. Afterward, solid ammonium sulfate was added to the supernatant slowly until the saturation reached 30% and after mixing, the solution was stood in a refrigerator at 4 °C for 6 h. After centrifuging at 14,000× *g* for 30 min, solid ammonium sulfate was added again to the supernatant until its saturation reached 40% and kept it at 4 °C for 4 h to precipitate. The mixture was then centrifuged under the same conditions and the precipitates were washed with 40% ammonium sulfate solution. Next, the precipitates were redissolved with 0.01 M phosphate buffered saline (PBS) (pH 7.4) and the solution was subsequently boiled for 10 min to denature some other proteins (precipitate was removed by centrifugation at 14,000× *g* for 10 min). At last, the supernatant was dialyzed for 24 h with 0.01 M PBS (10 kDa dialysis membrane was used) and PBS was changed 3 times. The final protein extracts were subjected to electrophoresis for identification. TM solution was adjusted to 0.5 mg/mL with 0.01 M PBS and stored at 4 °C until use.

### 2.3. Cold Plasma and Glycation Treatment

TM solutions were adjusted to 0.2 mg/mL with 0.01 M PBS (concentrations of protein were determined by Bradford Protein Assay Kit (Beyotime Biotechnology Co., Ltd., Shanghai, China), and BSA was used as standard) and then the ribose was added at a ratio of 1:4 (w/w) to protein, followed by magnetic stirring for 1 h in 25 °C. Next, 10 mL mixed solutions were exposed to dielectric barrier discharge (DBD) CP (Suman Co., Ltd., Nanjing, China) for 0, 1-, 2-, 3-, and 4-min time interval treatments, which would obtain CP treated TM (CTM). The treatment condition of CP was the same as our previous study [37]. Specifically, the plasma was produced at 60 kV, 1.0 A, and 10 mm discharge distance. After standing at room temperature for 1 h, the solution (adjusted pH to 7.8 with 1 M NaOH) was incubated at 80 °C for 4 h in the dry bath incubator (MAULAB Co., Ltd., China), which would cause glycation and obtain TM treated by CP combined with glycation (TTM) and glycated TM (GTM). Untreated TM was used as control. Finally, the samples were freeze-dried using SCIENTZ-18N freeze-dryer (Ningbo Scientz Biotechnology Co., Ltd., Ningbo, China) and stored at 4 °C until further use.

### 2.4. SDS-PAGE Analysis

The purified and treated TM (adjusted to 0.2 mg/mL with 0.01 M PBS, and 10 μL per lane) were analyzed by SDS-PAGE using precast gels. Electrophoresis was performed at 180 V for 50 min by NuPAGE Gel System (Invitrogen™, Thermo Fisher Scientific Inc., Carlsbad, CA, USA). The gel was placed into Coomassie Brilliant Blue R-250 (Beyotime Biotechnology Co., Ltd., Shanghai, China) stain solution for 1.5 h and then decolorized in water until the solution clear. Finally, the gel was archived by Bio-Rad gel imaging system (Model Gel Doc XR+, Hercules, CA, USA). All SDS-PAGE were imposed in reducing condition.

### 2.5. Antigenicity Analysis

#### Indirect Enzyme Linked Immune Sorbent Assay (ELISA)

The method previously described by Ekezie et al. [32] was used with some modifications. The concentrations of both samples were first adjusted to 20 μg/mL using 50 mM carbonate buffer (pH 9.6) and then 100 μL of above samples were placed overnight at 4 °C in the polystyrene 96 well plates (Xiamen YJM Labware Co. Ltd., Xiamen, China). Subsequently, the plates were washed 5 times with 200 μL PBST (0.01 M PBS, pH 7.4, 0.5% Tween-20) and blocked with 5% BSA (150 μL per well) at 37 °C for 2 h. After washing again with PBST, 100 μL/well human sera (1:20 dilution) were added to the plates and incubated at 37 °C for 2 h. After washing, the plates were incubated at 37 °C for 1.5 h with 100 μL/well anti-human IgE (1:5000 dilution in PBS). Then they were washed again, added 100 μL/well TMB was added and they were incubated at 37 °C for 10 min. Finally, 2 M sulfuric acid (50 μL/well) was used to stop the reaction and the OD_450_ was measured by a microplate reader (Thermo3001, Thermo Fisher Scientific Inc., Waltham, MA, USA).

### 2.6. Free Amino Group Content and Glycation Degree

The total free amino group content was determined by the o-phthalaldehyde (OPA) method as described by Zhao et al. [38] with some modifications. In brief, 3.81 g Na_2_B_4_O_7_·10H_2_O were dissolved in 75 mL distilled water, and then 0.1 g SDS and 0.088 g DTT were added. Then, 80 mg OPA was dissolved in 2 mL ethanol before being added into above solution. After adding OPA, the solution was kept away from light and was filled to 100 mL with distilled water in a brown volumetric bottle. Next, 200 μL of samples (0.2 mg/mL) were mixed with 4 mL OPA solution. Subsequently, it was incubated at 35 °C for 2 min, and then UV-Vis-NIR spectrophotometer (UV-1800, Shimadzu Co., Kyoto, Japan) was used to measure OD_340_.

The glycation degree was calculated according to the following formula:(1)$$\mathrm{Glycation}\;\mathrm{degree}=\frac{A_{0}{\;-\;A}_{1}}{A_{0}}\times 100\%$$
where A_0_ is the absorbance of the TM without glycation treatment at 340 nm and A_1_ is the absorbance of TM with glycation treatment at 340 nm.

### 2.7. Conformation Changes

#### 2.7.1. UV Absorption

Protein solutions (0.2 mg/mL) were scanned within the spectral range of 230–340 nm using a UV-Vis-NIR spectrometer. The spectra of the samples were acquired repeatedly: three times.

#### 2.7.2. Intrinsic Fluorescence

The intrinsic fluorescence emission spectra were obtained via a fluorescence spectrophotometer (RF-6000, Shimadzu Co., Kyoto, Japan). Samples (1 mg/mL) were scanned at an excitation wavelength of 280 nm and between 290 and 400 nm for emission. All spectra were acquired repeatedly (three times) with 5 nm constant slit width.

#### 2.7.3. Surface Hydrophobicity

According to Jin et al. [17], 8-aniline-1-naphthalene sulphonic acid (ANS) was used as a fluorescence probe to determine the surface hydrophobicity (H_0_) of protein solutions. The samples were diluted in series (0.4, 0.2, 0.1, 0.05, and 0.025 mg/mL) with 0.01M PBS. Subsequently, 2 mL of samples were mixed with 20 μL 8.0 mM ANS for 10 min, and kept away from light. Lastly, fluorescence intensity was recorded at an excitation wavelength of 390 nm and between 400 and 600 nm for emission via a fluorescence spectrophotometer. The slope of the plot of 470 nm (emission) fluorescence intensity against protein concentration, as derived by linear regression, was used as an index of H_0_. Measurements were performed in triplicate.

#### 2.7.4. Circular Dichroism (CD) Spectroscopy

The far CD spectra of protein solutions (adjusted to 0.1 mg/mL with 0.01M PBS) were acquired with a CD spectrometer (JASCO Co., Tokyo, Japan) at the spectral region from 190 to 260 nm at 25 °C. The step and bandwidth of the scan were all set to 1 nm. The content of secondary structures was calculated with CDNN software (Borland Software Co., Scotts Valley, CA, USA).

#### 2.7.5. Atomic Force Microscope (AFM) Analysis

The AFM (Park XE-70, Park Systems Inc., Suwon, Republic of Korea) analysis was conducted as described by Han et al. [26] with some modifications. Briefly, the samples (adjusted to 5 μg/mL) were dropped on a fresh mica (10 × 10 mm) surface and then air-dried in a fume hood for 6 h. Subsequently, the samples were observed in non-contact mode with scan rate and scope of 1 Hz and 2 × 2 μm, respectively. The images were processed and analyzed by XE Image Processing Program 5.0.1 (Park Systems Corp., Suwon, Republic of Korea).

### 2.8. Statistical Analysis

All experiments were performed in triplicates. Data were expressed as mean ± standard deviation (SD). The statistical software SPSS (version 19.0, SPSS Inc., Chicago, IL, USA) was used for a one-way analysis of variance (ANOVA), and the significant difference between the mean values was *p* < 0.05. Origin 2021 was used for data processing and analysis.

## 3. Results and Discussion

### 3.1. SDS-PAGE Analysis of Non-Treated and Treated TM

SDS-PAGE analysis can characterize the weight of protein to a certain extent, which is relevant to the modification of amino acid side chains during CP and glycation treatment [28]. As shown in Figure 1 the MW of native TM was about 34 kDa. The results showed that after CP treated alone, there was no visible change of CTM band. In addition, no new bands emerged at any treatment times. This indicated that TM had not been degraded or cross-linked after CP treatment. Similar results were obtained by Ekezie et al. [32], who treated TM of *Litopenaeus vannamei* with cold argon-plasma jet. However, the band density of TM from *Litopenaeus vannamei* decreased under non-reducing SDS-PAGE, which may be due to the interaction between proteins. In Figure 1B, compared with CP treated alone (C1–C4), MW of GTM was increased after glycation, which revealed the occurrence of glycation. Meanwhile, when CP combined with glycation processed TM, its MW increased slightly with the extension of CP treatment time. This may result from the protein unfolding induced by CP promoting the glycation. However, this slight movement of bands could also result from the uneven current during the electrophoresis. Therefore, compared with MW of TM after glycation, MW of TM did not change remarkably after combined processing treatment. In contrast, Yu et al. [34] observed that CP pretreatment could promote the glycation between peanut protein and lactose. This also indicated that there was some combined effect between CP and glycation on modifying TM. Due to the suffering from oxidation of CP and glycation, side-chain groups of TM were modified and these would affect linear and conformational epitopes of TM, eventually changing allergenicity of TM.

### 3.2. IgE Binding Ability Changes of TM Treated by CP and CP Combined with Glycation

The allergenicity response of TM was investigated via indirect ELISA. As shown in Figure 2, similar to previous studies [31], the allergenicity of TM did not decrease significantly after CP treatment alone. Even at a maximum CP treatment duration of 4 min, it only showed a 5% reduction. In a recent study, Ekezie et al. [32] extended CP treatment time to 12 min, but there was still no marked reduction in IgE binding ability of TM of shrimp. However, it was confirmed that CP treatment could reduce the allergenicity of wheat and soybean allergens effectively [27,28], which may be attributed to the nature of the allergens and the discharge mode of CP. Similarly, sole glycation treatment did not show a significant reduction in the allergenicity of TM (5% reduction). This was related to the reaction conditions of glycation, including reaction time, temperature, type of reducing sugar, and state of reactants [14]. It is worth nothing that heating alone (60 °C,12 h) also produced litttle reduction in IgE-binding capacity of TM [14]. However, once TM was pretreated with CP before glycation, its ability to bind with IgE decreased with the CP treatment time. It reached a 40% reduction at maximum treatment time. Hence, from this result, it could be deduced that the physical and chemical effect of CP could unfold or aggregate TM and expose new sites for glycation reaction. In this way, more linear epitopes were modified, resulting in a significant decrease in its allergenicity. Similarly, Yang et al. [39] reported that compared with the MR under atmospheric pressure, MR under high pressure could more effectively reduce the immunoreactivity of fish parvalbumin (PV). This could be attributed to the compression and remodeling of the protein under high pressure, which increased the exposure of the antigenic epitopes. Previous studies predicted that Lys was one of the key amino acids in IgE epitopes of prawn by many immunoinformatics tools (such as DNAStar Protean) [40]. Furthermore, Zhang et al. [41] explored the sites of glycation on TM of *Exopalaemon modestus* by quadrupole time-of-flight (Q-TOF) mass spectrometry (MS) and found that some glycation sites, particularly Lys (K_48_, K_149_, K_161_, K_248_, and so on) that were also the compositions of IgE epitopes, played a significant role in allergenicity reduction. Meanwhile, Lys would also be attacked by ROS and was the third priority, while aromatic and sulfur-containing amino acids had a higher priority [42]. For sequence of TM from *Penaeus chinensis*, there are 20 Lys (https://www.ncbi.nlm.nih.gov/protein/ADA70137.1, accessed on 1 March 2022) and around half of them were parts of predicted IgE epitopes [40,43]. Thus, in current study, the effective reduction in allergenicity was likely due to more modified Lys.

### 3.3. Changes in Free Amino Group Content and Glycation Degree of TM

As shown in Figure 3A, free amino group content of TM significantly dropped after glycation treatment, due to the generation of Schiff base, repeatedly revealing that the glycation had happened [44]. CP treatment also contributed to the decrease of free amino group content, resulting from the oxidation of amino groups by radicals, ROS, and RNS [42]. Meanwhile, the content of free amino group of TM under CP treatment was lower than it was under glycation treatment. Moreover, after combined treatment, free amino group content was further reduced. This suggested that, compared with glycation treatment alone, there were more free amino groups being modified after combined treatment. However, the degree of glycation was basically unchanged after 1 min and 2 min of CP treatment, but it decreased after 3 min and 4 min of CP treatment (Figure 3B). This indicated that the CP combined with glycation did not increase the extent of glycation, and the free amino groups did not react to produce more Schiff bases. Instead of this, it could be deduced that the oxidation of CP firstly modified the free amino groups with low steric hindrance, which preempted the sites of glycation [45]. Thus, in subsequent processing of glycation, ribose had to search for other free amino groups to react with, which needed to overcome larger steric hindrances. Nonetheless, it encouraged more amino groups to be modified, which meant that more potential antigen epitopes would be modified, leading to a further decrease in allergenicity of TM [46]. This deduction could also explain the result of immune response which is described above. Zhu et al. [47] also assessed the degree of MR after induction of free radical and the results showed that the free amino group content of myofibrillar proteins significantly decreased with the increase of ROO· concentration.

### 3.4. Effects of CP and CP Combined with Glycation Treatment on Conformational Stability of TM

#### 3.4.1. Effects of CP and CP Combined with Glycation Treatment on the UV Absorption of TM

UV absorption spectra can help to explore the conformational changes of protein [48]. The region around 280 nm is related to aromatic amino acids, including Phe, Trp, and Tyr [49]. As shown in Figure 4A, the increase of UV absorption intensity was observed after treatment. It indicated that the aromatic amino acid was exposed or modified after treatment. Takai et al. [42] reported that all 20 amino acids would be modified due to CP treatment, especially hydroxylation and nitration of aromatic rings in Phe, Trp, and Tyr. The absorbance of TM treated with CP combined glycation was higher than that treated with CP alone. Firstly, the presence of sugar group increased its absorption to UV [50]. Secondly, the thermal effect led to the exposure of aromatic amino acids [51]. Conversely, Meng et al. [52] observed that the UV absorption intensity of bovine β-lactoglobulin decreased with the increase of high hydrostatic pressure. This could result from the masking of aromatic amino acids and it would lead to irregular increase or decrease in allergenicity.

#### 3.4.2. Effects of CP and CP Combined with Glycation on the Tertiary Structure of TM

Aromatic amino acids have fluorescent characteristics, which are related to their microenvironment [49]. The intrinsic fluorescence intensity reflects the number of exposed aromatic amino acid residues in the aqueous phase [53]. Therefore, intrinsic fluorescence spectrum can reflect its conformational change. The results are shown in Figure 4B. It could be observed that the maximum absorption wavelength (λ_max_) was about 304 nm. This was because TM of *Penaeus chinensis* had no Trp, and its internal fluorescence characteristics were mainly determined by Tyr and Phe (https://www.ncbi.nlm.nih.gov/protein/ADA70137.1, accessed on 1 March 2022) [54]. Furthermore, whether CP or glycation treatment, the fluorescence intensity decreased and λ_max_ redshifted slightly (from 303 nm to 307 nm). This indicated that they all changed the microenvironment of TM, which would influence hydrogen bonding, van der Waals, and hydrophobic and electrostatic interaction [26]. Previous study showed that the unfolding structure of protein was related to the changes in polarity environment [55]. Hence, it could be inferred that the tertiary structure of TM changed after CP or glycation treatment. In comparison with separate treatment, combined treatment further reduced the fluorescence intensity. This may be because pretreatment of CP promoted the covalent binding between Tyr to ribose [34]. Other studies also suggested that physical or chemical processing of proteins before glycation would further destroy their tertiary structure. It was reported that oxidation before glycation destroyed the tertiary structure of myofibrillar proteins, which depended on the concentration of oxidant [47]. Yang et al. [56] also found that intrinsic fluorescence intensity of β-lactoglobulin decreased significantly after pretreatment of ultrasound and then glycation, revealing the more disruptive changes in conformational structure of β-lactoglobulin.

Furthermore, the changes of surface hydrophobicity can help to explain the conformational changes. As shown in Figure 4C, surface hydrophobicity increased firstly and then decreased, with the figure reaching the maximum at 2 min of CP treatment. The increase of surface hydrophobicity indicated that more hydrophobic groups were exposed, which was consistent with the downward trend of internal fluorescence intensity. Its reduction in subsequent treatment (>2 min) may be due to the re-association or aggregation of TM. Ekezie et al. [57] found similar results using cold plasma jet to treat actomyosin of shrimp. Jin et al. [17] treated TM of squid with high hydrostatic pressure and also found that its surface hydrophobicity firstly increased and then decreased. Meanwhile, its surface hydrophobicity was higher under cooperative treatment. This was because glycation caused the structure of protein to unfold further, exposing the hydrophobic region to the outside of the protein molecule.

#### 3.4.3. Effects of CP and CP Combined with Glycation on the Secondary Structure of TM

The winding and folding of peptide chains constitute the secondary structure of proteins and the form includes α-helix, β-sheet, β-turns, and random coils [58]. CD spectrum is the most common method to analyze the secondary structure of proteins [59]. As shown in Figure 4D, the native TM of shrimp was equipped with the obvious α-helix structure (a positive peak at around 196 nm and two negative peaks at 208 nm and 222 nm). When it was experienced with glycation or CP treatment, the α-helix characteristic peak intensity decreased and converted to β-turns and random coils (Figure 4E). It was worth noting that a small content of β-sheet structure was generated after glycation. Moreover, a large number of random coils structures were generated when samples were exposed to combined treatment. The changes in the secondary structure of protein illustrated that TM was constantly unfolded and reorganized under the treatment. Due to the heat stability of TM, thermal treatment could not change its secondary structure [14]. However, in some extremely thermal conditions, it can lead to a 10% reduction of α-helix of TM [51]. Furthermore, Venkataratnam et al. [30] reported that the secondary structure of Ara h1 which was the main allergen of peanuts showed a dramatic variation after CP treatment, while the random coil of it did not change significantly. This may be due to its own irregular structure (35% of random coil). Lv et al. [54] also found that α-helix of TM reduced after being glycated with ribose. Previous studies showed that the transition from ordered structure to disorder was beneficial to masking or obscuring antigenic epitopes [60]. As the support for the tertiary structure, secondary structure of protein plays an important role in maintaining conformational stability of TM, which would affect its allergenicity. The oxidation of CP and glycation with ribose rearranged the secondary structure of TM, leading to the expansion or aggregation of proteins. This influenced its conformation and possibly contributed to reducing its binding ability to IgE.

#### 3.4.4. Effects of CP and CP Combined with Glycation on the Surface Morphology of TM

As depicted in Figure 5, AFM was employed to obverse the morphology, roughness, and height distribution of TM particles. 3D images and root mean square roughness (RMS) were obtained from XEI 5.0.1. For TM treated with CP (Figure 5A), it was dispersed obviously by the first 2 min of CP treatment. The RMS reduced from 0.814 nm to 0.529 nm. Subsequently, small protein particles aggregated to form larger particles with the increase of RMS from 0.529 nm to 5.611 nm. At 4 min of CP treatment, the larger aggregates particles were generated and maximum particle height could reach 30 nm. It indicated that the oxidation of CP made molecules of TM disperse first and then aggregate. This trend was consistent with that of surface hydrophobicity. Similar trends of aggregation happened to HRP treated by CP jet [26]. Zhu et al. [61] also found that polyphenol oxidase (PPO) of mushrooms aggregated directly after CP treatment. This was probably because the native PPO particles themselves were too small to disperse. In contrast, TM treated by glycation tended to form round aggregates (Figure 5Bi). This may be due to changes in the morphology of the molecule caused by the sugar group. Similar results were obtained by Xu et al. [62], who observed non-uniformed globules when myofibrillar protein was glycated with dextran. Furthermore, under the combined effect of CP and glycation, the aggregation degree of TM particles did not change significantly and RMS fluctuated around 2 nm, while the particles tended to be circular. This may be attributed to the dispersion of CP and the thermal effect during glycation, which led to a trend of outward diffusion of particles. According to the change of RMS (Figure 5C), it can be inferred that glycation can disperse the assembled molecules which were aggregated by CP. This had an important effect on its conformation and may be one of the reasons for the reduction in allergenicity.

## 4. Conclusions

The combined effects of CP with glycation on allergenicity and conformation of TM were investigated in this study. The results showed that only a slight decrease in allergenicity of TM occurred after CP or glycation treatment. However, a significant reduction in allergenicity occurred when it was induced by CP combined with glycation. The immunoreactivity of TM reduced by 40% after CP treatment for 4 min and then glycation. The treatment time of CP had a pronounced influence on the combination effect. With the combined treatment, the free amino content and the glycation degree of TM decreased, while the MW of TM did not change markedly. Based on these results, it was assumed that the oxidation effect of CP first modified the free amino groups with low steric hindrances and subsequently led ribose to search for other free amino groups with larger steric hindrances to react with. Moreover, the conformation of TM changed remarkably after the combination of CP with glycation treatment, including exposure and polarity shift of aromatic amino acids, increase of surface hydrophobicity, decrease of proportion of α-helix, and increase of proportion of random coil. Morphology analysis of TM also suggested the occurrence of aggregation of TM after a long time of plasma or glycation exposure. However, the co-processing made the distribution of TM particles tend to disperse circularly. These modifications in conformation may collectively change the IgE-binding capacity of TM. Therefore, the method of CP pretreatment combined with glycation has great potential to modify the structure and reduce or eliminate allergenicity of seafood.

## Figures and Tables

**Figure 1 foods-12-00206-f001:**
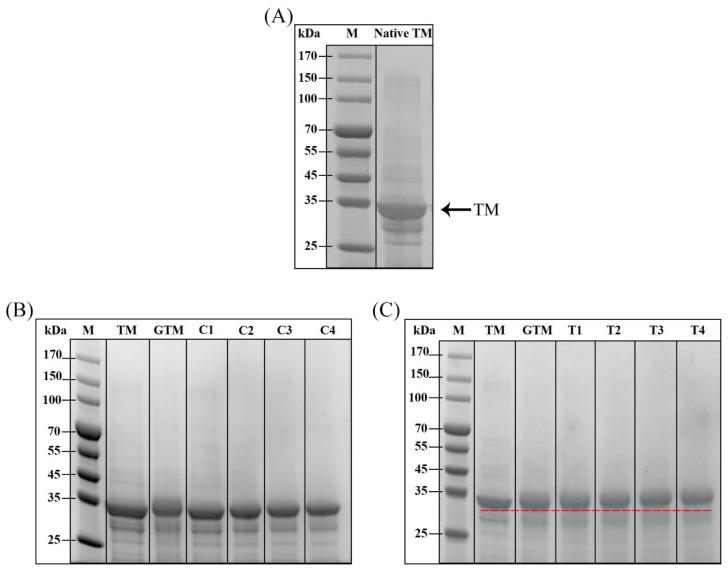
SDS-PAGE profiles of untreated and treated TM. (**A**) Extracted TM. (**B**) TM treated with glycation and CP, respectively. (**C**) TM treated by glycation alone and CP combined with glycation. M: Protein marker; TM: Native TM (control); GTM: TM treat with glycation alone; C1–C4: TM treated by CP for 1, 2, 3, 4 min, respectively. T1–T4: TM treated by CP for 1–4 min and then combined with glycation. The legends of the figure below were the same. Red dashed line was used as auxiliary line to help differentiate weight of proteins.

**Figure 2 foods-12-00206-f002:**
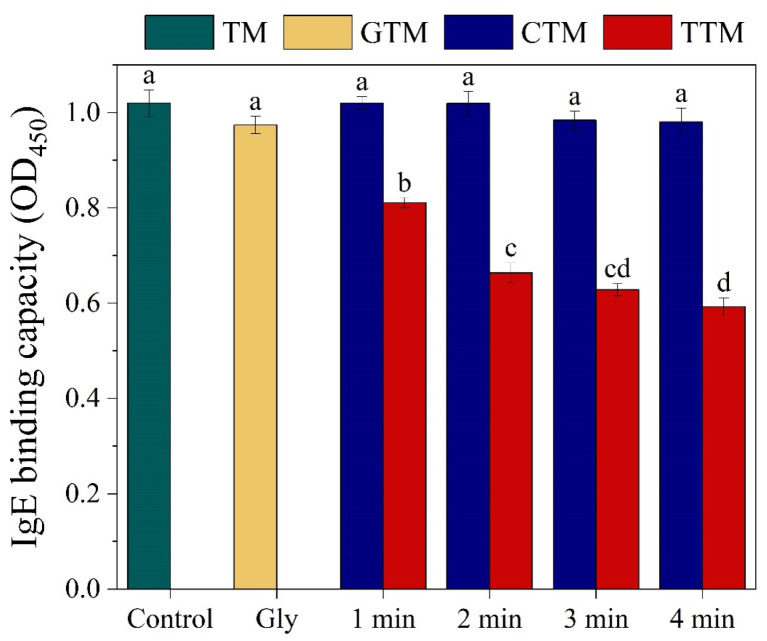
Indirect ELISA response of TM treated by CP, glycation, and CP combined with glycation against IgE binding activity using shrimp patient serum. TM: untreated TM. GTM: glycated TM. CTM: TM treated with CP. TTM: TM treated with glycation and CP. The legends of the figure below were the same. Values were means ± SD. Different letters indicate significant differences (*p* < 0.05).

**Figure 3 foods-12-00206-f003:**
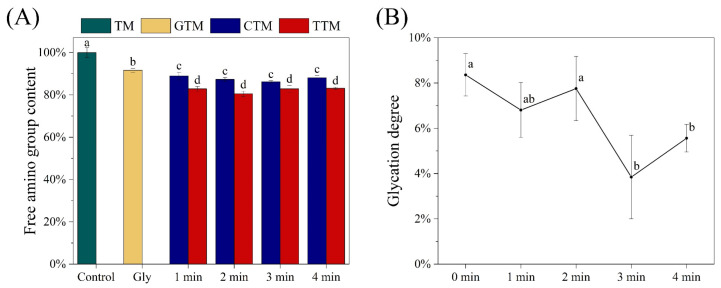
Changes in free amino content (**A**) and glycation degree (**B**) of treated TM. Values were means ± SD. Different letters indicate significant differences (*p* < 0.05).

**Figure 4 foods-12-00206-f004:**
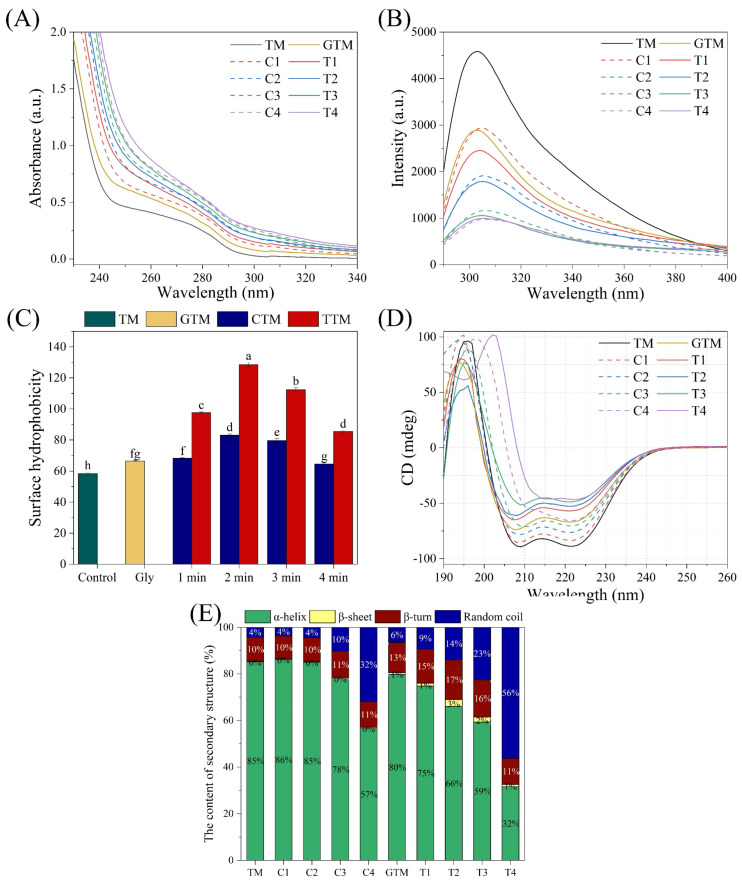
Effects of CP and CP combined glycation treatment on the UV absorption (**A**), intrinsic fluorescence (**B**), surface hydrophobicity (**C**), circular dichroism spectra (**D**), and secondary structure content (**E**) of TM. Different letters indicate significant differences (*p* < 0.05).

**Figure 5 foods-12-00206-f005:**
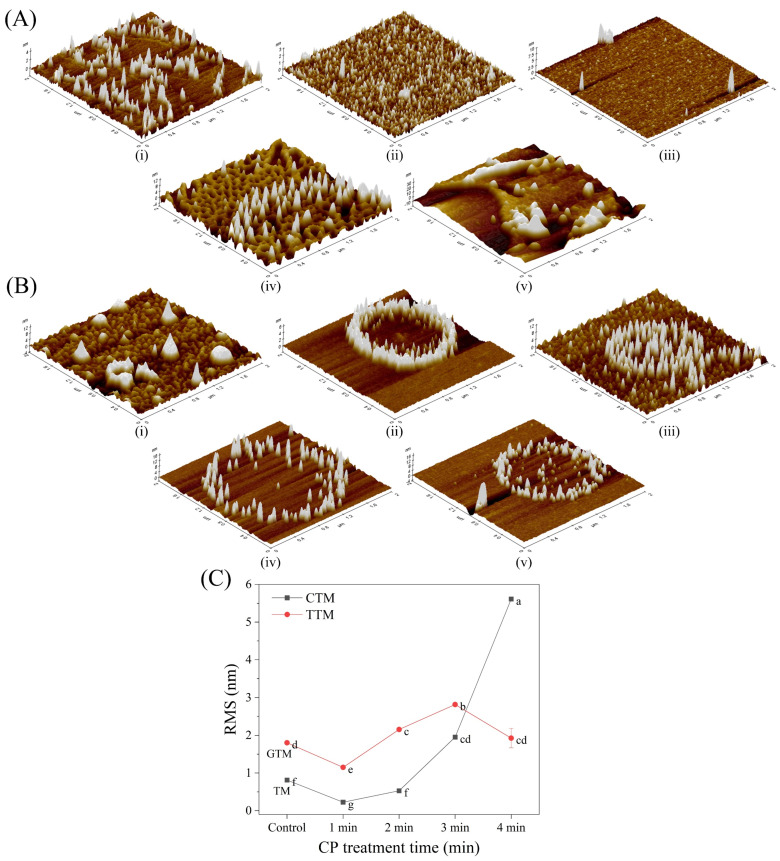
Surface morphology of TM treated by CP alone (**A**) and treated by CP combined with glycation (**B**), and their changes of RMS (**C**). (**i**–**v**): TM treated by CP for 0, 1, 2, 3, 4 min, respectively. Values were means ± SD. Different letters indicate significant differences (*p* < 0.05).

**Table 1 foods-12-00206-t001:** Serological characterization of shrimp allergy patients.

No.	Sex ^a^	Age (Years)	Shrimp IgE Level (kU/L) ^b^
P1	M	33	64.1
P2	F	38	24.5
P3	M	52	60.1
P4	M	61	35.0
P5	M	31	26.7
P6	F	21	22.1
P7	F	27	17.5
P8	F	29	20.1

^a^ M, male; F, female. ^b^ A serum with specific IgE > 0.35 (kU/L) is defined as positive.

## Data Availability

The data presented in this study are available on request from the corresponding author.

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
