# Peer review of "Changing the IgE Binding Capacity of Tropomyosin in Shrimp through Structural Modification Induced by Cold Plasma and Glycation Treatment"

_foods, 2023, doi:10.3390/foods12010206_

Round 1

Reviewer 1 Report (Previous Reviewer 2)

The manuscript has improved upon the previous submission. I do not have other criticisms.

Author Response

Dear reviewer,

Thank you very much for your comments and professional advice. These opinions help to improve academic rigor of our article. Based on your suggestions and requests, we have made corrected modifications on the revised manuscript. We hope that our work can be improved again. Furthermore, we would like to show the details as follows:

  1. The manuscript has improved upon the previous submission. I do not have other criticisms. Extensive editing of English language and style required.

The author’s answer: Thank you for your valuable suggestions. We have revised in the article (line 14, 15-17, 18, 19, 23-24, 25, 27, 31-35, 37, 39-40, 44, 48, 58-59, 60, 65, 68-69, 76-78, 108, 118, 159-160, 190-191, 246-247, 249, 252-253, 288-289, 291, 293, 296-297, 300, 302, 318-319, 321-322, 329, 345-346, 348, 349-350, 354, 356, 357, 362, 379-380, 384-385, 386, 397, 399, 402, 404, 405, 407-408, 429-430, 435). 

Reviewer 2 Report (Previous Reviewer 3)

Dear Authors,

Sorry but I still have no confidence in the method you use for the degree of hydrolysis. The methods are different, not only because of the OPA or ninhydrin; please read (Panasiuk et al., Food Chemistry, Vol. 62, No. 3, pp. 363-367, 1998). “Slight modification,” as you cold it, needs standardization. Please read a little about how others are doing that, Hu et al. Journal of Analytical Methods in Chemistry Volume 2014, Article ID 728068, 7 pages http://dx.doi.org/10.1155/2014/728068.

Few more concerns:

L40: glycation is a spontaneous reaction; Maillard reaction is glycation; If you would like to mention glycation products, please rewrite the sentence

L45: Glycation is a spontaneous reaction. This process occurs even if we do not plan to treat protein by glycation. Please rewrite.

L71: the extent of glycation

L89: molecular weight protein standard

L93: positive for allergy?

L95: Does it mean that all sera were mixed and used as one sample? Please correct.

L113: The boiling process is not for removing. The precipitate was removed by….. What was dialyzed? Please correct.

L165: Is a TM sample without glycation termized in the same conditions as glycated TM?

L209: I have to repeat: the protein's primary structure is the protein amino acid sequence. Glycation does not change it. Glycation alters only molecular weight, but it is not visible in 1D SDS-PAGE. Please correct.

L230: The same as below????? Red dashes do not help in anything. It is visible that different amounts of protein were loaded on the left and right gels (the bands' darkness is different). This “slight” 1 mm difference could be just an effect of electrophoresis; look for M lines on both gels. Please rewrite.

L261, 286: “Values in the same column with different letters were significantly different.” I see one letter with one column. What is this description about?

L374: There is even no column on the plot.

Please read the whole manuscript carefully and correct it. The method for the determination of DH needs to be strongly proven.

Author Response

Dear reviewer,

Thank you very much for your comments and professional advice. These opinions help to improve academic rigor of our article. Based on your suggestions and requests, we have made corrected modifications on the revised manuscript. We hope that our work can be improved again. Furthermore, we would like to show the details as follows:

  1. Sorry but I still have no confidence in the method you use for the degree of hydrolysis. The methods are different, not only because of the OPA or ninhydrin; please read (Panasiuk et al., Food Chemistry, Vol. 62, No. 3, pp. 363-367, 1998). “Slight modification,” as you cold it, needs standardization. Please read a little about how others are doing that, Hu et al. Journal of Analytical Methods in Chemistry Volume 2014, Article ID 728068, 7 pages http://dx.doi.org/10.1155/2014/728068.

The author’s answer: Thank you for your constructive guidance about the difference between methods of OPA and ninhydrin. But for the method of OPA we used, we think it has enough reliability because a large number of recent studies used similar methods of OPA (https://doi.org/10.1016/j.ijbiomac.2020.04.255, https://doi.org/10.1016/j.foodchem.2022.133215, https://doi.org/10.1021/acs.jafc.1c03953 , https://doi.org/10.1016/j.ultsonch.2020.104964 , and so on). And we did not wait for the solution to cool before measuring OD340. What’s more, we also conducted the standard curve with the L-Leucine (R2=0.9898), which showed that the methods of OPA was reliable. However, we think the description in this part maybe not nonstandard and have revised the description (line 165-170).

  1. L40: glycation is a spontaneous reaction; Maillard reaction is glycation; If you would like to mention glycation products, please rewrite the sentence. L45: Glycation is a spontaneous reaction. This process occurs even if we do not plan to treat protein by glycation. Please rewrite.

The author’s answer: Thank you for your correction about Maillard reaction and glycation. We have revised in the article (line 41-43).

  1. L71: the extent of glycation

The author’s answer: We have revised in the article (Line 77).

  1. L89: molecular weight protein standard

The author’s answer: We have revised in the article (Line 95).

  1. L93: positive for allergy?

The author’s answer: We have revised in the article (Line 99).

  1. L95: Does it mean that all sera were mixed and used as one sample? Please correct.

The author’s answer: We have revised in the article (Line 102).

  1. L113: The boiling process is not for removing. The precipitate was removed by….. What was dialyzed? Please correct.

The author’s answer: Thank you for your correction. We have revised in the article (Line 120).

  1. L165: Is a TM sample without glycation termized in the same conditions as glycated TM?

The author’s answer: Except the glycation treatment, other conditions of TM samples in experiments of Free amino group content are identical.

  1. L209: I have to repeat: the protein's primary structure is the protein amino acid sequence. Glycation does not change it. Glycation alters only molecular weight, but it is not visible in 1D SDS-PAGE. Please correct.

The author’s answer: Thank you for your patient correction. We have corrected our previously wrong concept and have revised in the article (line 216-218, 234-236). For molecular weight, it was visible in Figure 1B and 1C that the molecular weight of TM (control) and GTM (only glycation) is different. But for molecular weight of GTM and T1-T4 (CP combined with glycation), we have to admit that it may be not evident through Figure 1C, with slight upward movement. Therefore, we made some revision to become more rigorous (line 228-231, 424).

However, the difference really exists, which cannot be neglected. We think this unclear difference of molecular weight result from the mild reaction condition of glycation, including reaction time and monosaccharide. In recent experiments, we found it is true that molecular weight of TM sample treated with CP and glycation higher than it of others, and difference is clear in the Western Blot results (shown in S1, which have not been published).

S1. The results of Western Blotting of treated TM. M: Protein marker; TM: Native TM (control); G: TM treat with Gly alone; C1+G-C4+G: TM treated by CP for 1-4 min and then combined with Gly.

  1. L230: The same as below????? Red dashes do not help in anything. It is visible that different amounts of protein were loaded on the left and right gels (the bands' darkness is different). This “slight” 1 mm difference could be just an effect of electrophoresis; look for M lines on both gels. Please rewrite.

The author’s answer: We have revised in the article (line 242). We think that the difference of darkness can be attributed to the exposure time of camera. Because the M line (molecular weight protein standard) is also darker on the left gel, but the marker in the left gel definitely has the same amount with the right one. And we actually conducted with the same amounts of samples in the both gels.

For the slight difference of Figure 1C, it is true that it remains controversial so we make some revision (line 228-231, 424). But the slight increase of molecular weight existed (as shown in the figure above). Thank you for your worthwhile and rigorous suggestion.

  1. L261, 286: “Values in the same column with different letters were significantly different.” I see one letter with one column. What is this description about? L374: There is even no colum1.

The author’s answer: Thank you for your careful correction. We have revised in the article (Line 281, 307, 327, 393). 

Reviewer 3 Report (New Reviewer)

The present work focuses on the application of cold plasma and glycation as a potential processing method to reduce tropomyosin allergenicity. The results are promising since the authors could reach a 40% reduction in IgE-binding to tropomyosin, which could be a viable solution for the production of hypoallergenic formulas for shrimp allergic patients. I think that this manuscript needs some substantial improvement of English language. Therefore, I suggest the authors to search for help within native English experts.

Besides this point, some minor corrections should be addressed:

The introduction can be improved. The authors can explore more tropomyosin as an allergen and the effect of processing as a strategy to reduce proteins allergenicity.

Lines 32-33. Confusing sentence.

Line 44. Change to “allergenic epitopes”.

Line 46. Change to “and increase allergenicity”.

Lines 56-57. Change to “Regarding the reduction of food allergenicity”.

Line 58. Change to “Shriver [30] found a 10% reduction”.

Line 60. Change to “serious”.

Lines 67-68. Confusing sentence. Rephrase.

Lines 107-109. Rephrase these sentences because they are written in the present as a protocol.

Line 216. To which proteins the authors are referring here?

Line 253. Change to “exposure”.

Lines 255-256. The reviewer is confused about the involvement of Lys in this work. Tropomyosin contains enough Lys to produce a measurable effect on IgE-binding because of glycation? The authors can explore more this topic.

Line 403. I guess that the authors means a reduction in 40% since reported in the abstract and in section 3.2.

Author Response

Dear reviewer,

Thank you very much for your comments and professional advice. These opinions help to improve academic rigor of our article. Based on your suggestions and requests, we have made corrected modifications on the revised manuscript. We hope that our work can be improved again. Furthermore, we would like to show the details as follows:

  1. The introduction can be improved. The authors can explore more tropomyosin as an allergen and the effect of processing as a strategy to reduce proteins allergenicity.

The author’s answer: Thank you for your constructive advice. We have made some revision in the article (line 49-55).

  1. Lines 32-33. Confusing sentence.

The author’s answer: We have revised in the article (line 32-33).

  1. Line 44. Change to “allergenic epitopes”.

The author’s answer: We have revised in the article (line 45).

  1. Line 46. Change to “and increase allergenicity”.

The author’s answer: We have revised in the article (line 46).

  1. Lines 56-57. Change to “Regarding the reduction of food allergenicity”.

The author’s answer: We have revised in the article (line 62).

  1. Line 58. Change to “Shriver [30] found a 10% reduction”.

The author’s answer: We have revised in the article (line 64).

  1. Line 60. Change to “serious”.

The author’s answer: We have revised in the article (line 67).

  1. Lines 67-68. Confusing sentence. Rephrase.

The author’s answer: We have revised in the article (line 73-75).

  1. Lines 107-109. Rephrase these sentences because they are written in the present as a protocol.

The author’s answer: We have revised in the article (line 113-116).

  1. Line 216. To which proteins the authors are referring here?

The author’s answer: We have revised in the article (Line 224-226).

  1. Line 253. Change to “exposure”.

The author’s answer: We have revised in the article (line 265).

  1. Lines 255-256. The reviewer is confused about the involvement of Lys in this work. Tropomyosin contains enough Lys to produce a measurable effect on IgE-binding because of glycation? The authors can explore more this topic.

The author’s answer: Thank you for your valuable suggestions. We have made some supplements in the article (line 266-275).

  1. Line 403. I guess that the authors means a reduction in 40% since reported in the abstract and in section 3.2.

The author’s answer: We have revised in the article (line 421).

Round 2

Reviewer 2 Report (Previous Reviewer 3)

Dear Authors,

I accepted most of all corrections.

Still, the OPA methods and results are in "blind point." Let me clarify the situation. You applied this manuscript earlier to Foods with the same title, and the manuscript got the number foods-1838362. In the review, I show you the factual errors concerning the OPA method from the Material and Methods chart with a point of 2.6. "Free amino content and grafting degree." After you corrected the manuscript and applied it again, with the same title to FOODS, now it has a number foods-2004650. The difference is that now you describe correctly and quote the OPA method properly. But you still do not give me any reason to trust those results. I am glad you finally familiarized with the correct methodology for the determination of free amine groups and the degree of hydrolysis:

https://doi.org/10.1016/j.ijbiomac.2020.04.255, https://doi.org/10.1016/j.foodchem.2022.133215, https://doi.org/10.1021/acs.jafc.1c03953 , https://doi.org/10.1016/j.ultsonch.2020.104964

I will leave it to the Editor's decide about this situation.

Other comments concerning glycation:

1)    Two sentences explaining the reason for the glycation (L284-285) could be used as a possible mechanism for improving your results. Written in form like now suggests the analysis done in presented experiments. Changing sentence order will resolve this solution.

2)    Figure 3A shows no significant differences in free amino groups between TTM samples after 1, 2, 3, and 4 min of CP treatment. Could you please explain why you have differences in glycation degree in samples after 3 and 4 min CP treatment? According to a formula you provided

we do not expect differences between them because:

·      A0 Is the same for all samples

·      A1 values between samples are without differences (Fig.3A),

The SD values for GlycDeg or each sample are subject to significant error, which suggests a large variance between replicates, a poorly performed method, and may be no differences. The sentence started. However, (L292) is not proven by statistical analysis, with no differences. Rethink this paragraph, please.

Author Response

This manuscript is a resubmission of an earlier submission. The following is a list of the peer review reports and author responses from that submission.

Round 1

Reviewer 1 Report

Wang and colleagues have studied the structural modifications of shrimp tropomyosin, induced by cold plasma and glycation treatment. They applied these treatments separately and in combination and assessed their impact on IgE-binding. Shrimp is a major allergen source and a reduction of the allergenicity of shrimps would be very beneficial to patients.

Whereas the structural analysis is well presented and the applied methods are sound, the part on IgE-binding is less convincing. In particular, IgE-binding as analysed in ELISA is only part of the allergenicity assessment of an allergen source. The shrimp allergic patient sera used are not characterized, they were purchased and no details are given on the clinical reactivity. An sIgE level > 0.35 seems to be the only inclusion criteria. How many sera were used and what are the sIgE values of those sera? Where these patients sensitized only or also clinically reactive?

How many times were the glycation and CP experiments repeated ? Are the results statistically significant?

There was a problem with the figure references in the text file. References are not always clear.

The study has several weaknesses which should be discussed. Purified tropomyosin was investigated, it is not known how the matrix effect of whole shrimp would impact the result. As IgE-binding is only part of an allergenicity assessment of food, more studies are necessary. Patient sera are not well characterized.

Figure 1: please specify if the gel was run under reducing conditions

Figures 2 and 3: how many sera were used to determine IgE-reactivity? Individual values should be represented as dots, and IQR given. A non-treated control with incubation for 4h at 80 °C is missing, as well as native, untreated TM, purified under mild conditions. The legend is not clear: what are the different a bars, what is b, c, and cd ? Same for figure 3. Are these different preparations?

Figure 4: please annotate y-axis with units

Lines 109-111: tropomyosin was purified by ammonium sulfate precipitation steps, followed by 10 min of boiling. As this step is a very harsh condition for a protein, authors should control IgE-reactivity of the protein before and after this step. Some denaturation may already have occurred during purification.

Line 116: what were the temperature conditions during incubation?

Line 121: 4h at 80 °C is a very stringent condition. The figures do not contain controls incubated under the same condition, but without CP treatment or ribose addition.

Line 138: please give the % of the BSA used

Line 178: please mention buffer used for the measurement

Line 298: please give the numbers for the redshift, it is not visible on the graph

Reviewer 2 Report

The authors evaluated the effect of CP and Gly treatments, alone or in combination, on the allergenicity of TM, the main allergen of shrimp.

General comments:

The data generated are well presented (all figures), described, and discussed, and the conclusions are supported by the results. I found the experiments well designed. I believe the manuscript adds significantly to our understanding of food allergies and food processing.

I have minor criticisms.

Specific comments.

Abstract:

1.-Please, provide p values when you compare the data and give the data (percentages) of single CP and Gly treatments (lines 16-17).

Introduction:

The introduction section goes from the general to the particular, and key information is highlighted.

Lines 40-41. Step instead of steps. Additionally, use instead of employed (this last suggestion is up to you).

Materials and methods:

Lines 109-111. Please briefly describe the principle of boiling to remove denatured protein. Also, provide data about the cut-off of the dialysis membrane.

Line 115. Please provide data about the method to estimate the TM concentration, including the protein used to build the standard curve.

Line 120. Consider the following: (adjusted pH WITH NaOH; use WITH instead of BY). Also, provide the molarity of the NaOH used to adjust the pH.

Lines 126-127. Provide data about how many microliters of the 0.2 mg/mL TM solution were charged per lane to carry out the SDS-PAGE analysis. You could also provide the data in micrograms of TM per lane.

Line 134. Please, conclude the sentence, perhaps, ….was used.

Line 141. Please provide data on the anti-human IgE in micrograms as well.

Results and discussion:

Line 203-204. Consider the following: And similar results were obtained by….. 

Line 222. Something appears to be wrong; please investigate.

Line 237. Please specify what IT refers to (…pressure, IT under high pressure…) 

Why was the IgE binding capacity using raw TM (untreated) as an antigen not presented? Or, does CP time 0 mean untreated? Please, clarify it. The same happens in Figure 3.

Lines 249-250 and 256. It seems that something was wrong. Please check it throughout the results and discussion section.

Conclusion.

 Lines 378-379. This sentence needs to be completed (The results showed that….). For instance, the results showed that only a slight decrease in THE allergenicity of TM OCCURS after CP or Gly treatment.

Reviewer 3 Report

Dear Authors

The first impression is that it is an interesting manuscript. It looks well planned. Then the errors start. It even seemed to me that you confused the determination of the concentration of free amino groups in peptides with free amino acids in solution. This is a serious factual error, and it's early on.

You write about the primary structure of proteins (amino acid sequence) as if it were molecular weight - another factual error.

The glycation process is a spontaneous reaction in which a reducing sugar attaches to free amino groups in a protein. The OPA method is used to determine the number of free amino groups in a protein and does not use ninhydrin.

I have included a few more comments below. The manuscript requires rethinking the concept of the experiments performed, correcting errors, and rewriting.

Round 2

Reviewer 1 Report

The authors have adressed most of the comments, but there is still some clarification needed regarding the allergic sera and experimental setup.

Sera have been mixed equally for Elisa. Does this mean that sera have been pooled for the assay? If so, this is an important point and a serious weakness of the study and needs to be addressed.

It is still also not clear if patients were really allergic to shrimp or only sensitized. Please confirm.

Arguments regarding the missing heat stability experiments are valid, but should be discussed in the results and discussion section.
